# Depressive and Anxiety Symptoms and Their Relationships with Ego-Resiliency and Life Satisfaction among Well-Educated, Young Polish Citizens during the COVID-19 Pandemic

**DOI:** 10.3390/ijerph191610364

**Published:** 2022-08-19

**Authors:** Agnieszka Goryczka, Paweł Dębski, Anna M. Gogola, Piotr Gorczyca, Magdalena Piegza

**Affiliations:** Department of Psychiatry, Faculty of Medical Sciences in Zabrze, Medical University of Silesia, 42-612 Tarnowskie Góry, Poland

**Keywords:** ego-resiliency, life satisfaction, depression, anxiety, COVID-19 pandemic

## Abstract

Ego-resiliency is a set of traits that promotes positive adaptation to life’s vicissitudes. High ego-resiliency helps in upholding one’s personality system when facing adversity and in adjusting it to new environmental demands. Our study aimed at evaluating the connections between ego-resiliency, the severity of anxiety and depressive symptoms as well as life satisfaction during the COVID-19 pandemic in Poland. A total of 604 Polish citizens aged 16 to 69 years participated in the online survey. Ego-resiliency was measured with the Ego Resiliency Scale (ER89-R12), anxiety and depression with the Hospital Anxiety and Depression Scale (HADS), and life satisfaction with the Satisfaction with Life Scale (SWLS). Statistical analyses were performed using the Spearman rank correlation coefficient. The results revealed correlations between the intensity of depressive and anxiety symptoms, life satisfaction, and the intensity of ego-resiliency. Individuals with a high level of ego-resiliency tended to experience a lower intensity of anxiety and depressive symptoms during the COVID-19 pandemic. Moreover, individuals with a high level of ego-resiliency exhibited a higher level of life satisfaction. Our conclusions might assist in better understanding the close link between levels of ego-resiliency, the occurrence of depressive and anxiety symptoms, and satisfaction with life among Polish individuals experiencing crises.

## 1. Introduction

Resiliency is one of the crucial coping mechanisms that promotes healthy and beneficial behaviors during crises [1,2,3]. It is defined as a dynamic ‘processual phenomenon’ that allows ‘positive adaptation’ to vicissitudes [4]. It enables human beings to maintain their personality systems in equilibrium when in a quickly changing environment. Unlike resiliency, the construct of ego-resiliency (ER), which was first described by Block [5], is not a processual phenomenon but rather a set of personality traits that promotes adaptation to situational demands (e.g., psychological stability and flexibility). It incorporates two important properties of the human psyche: stability and flexibility. Stability allows humans to uphold their personality systems when facing adversity, whereas flexibility signifies the ability to simultaneously adjust it (if necessary) to the new demands of the environment [6].

‘Ego-resilient’ individuals are not only good at adapting to new situations, but they are also able to see positive aspects in problems that they have to face. They are characterized by so-called ‘positive emotionality’ [1,7]. Additionally, they tend to be ambitious and extraverted [8,9]. Conversely, individuals who can be defined as ‘ego-brittle’ experience more anxiety [2]. They struggle with responding dynamically to changing conditions and tend towards perseveration [8]. Furthermore, ‘ego-brittle’ individuals are more likely to demonstrate high levels of neuroticism (it has been proven that resiliency has a strong negative connection with neuroticism [9]). Individuals who score high on neuroticism and low on extraversion might be more prone to depression [10]. From this we can conclude that ‘ego-brittle’ individuals are likely to have problems with successful adaptation and maintaining mental health. These observations are especially relevant due to the fact that our study was conducted during the COVID-19 pandemic, which altered the reality of everyday life. Pandemic restrictions were linked to elevated stress levels, anxiety and depression, and the prevalence of these disorders was high [11,12]. All these factors posed a threat to people’s mental health. Further research on homeostatic mechanisms and traits, which help individuals to bounce back from adversities, has become a priority in recent mental health studies. Therefore, it is important to keep investigating the role of factors that have a protective effect on people’s mental health, such as ego-resiliency.

During the COVID-19 pandemic, many studies measured life satisfaction and explored its fluctuations. Pandemic restrictions and elevated stress levels caused a decrease in life satisfaction [13,14]. A life satisfaction assessment was also included in our study in order to further investigate its relationship with ego-resiliency. The notion of life satisfaction can be controversial and is understood differently by various authors. We defined it as the ‘cognitive-judgmental component of subjective well-being’ [15], and our approach to this concept is consistent with the one proposed by the Life Satisfaction Theory. It assumes that well-being and the emotional state of being satisfied with one’s life are equivalent [16]. Recently, there has been a notable increase in research on well-being [16,17,18]. Therefore, it might be important to investigate factors that exhibit a correlation with life satisfaction, i.e., ER.

The aim of the study was to evaluate the associations between ego-resiliency and the severity of anxiety and depressive symptoms, as well as satisfaction with life during the COVID-19 pandemic in Poland. Moreover, the purpose of our analyses was also to compare people who declared no change, a positive change, or a negative change in their mental functioning during the COVID-19 pandemic.

## 2. Materials and Methods

### 2.1. Design and Participants

In our cross-sectional study ER, anxiety, depression, and satisfaction with life were measured using the convenience-sampling method. The online survey was created using Google Forms and consisted of a sociodemographic questionnaire and psychological scales: the Ego-resiliency Scale (ER89-R12), Hospital Anxiety and Depression Scale (HADS), and Satisfaction with Life Scale (SWLS). It was shared via social networks (Facebook and Instagram) from 10 November to 2 December 2020. The inclusion criteria were: age over 16 years, ability to use the computer and the internet, ability to write and read in Polish.

### 2.2. Measures

#### 2.2.1. Ego-Resiliency Scale (ER89-R12)

The level of ER was measured using the Polish adaptation of the Ego-Resiliency Scale (ER89-R12) developed by Kołodziej-Zaleska and Przybyła-Basista [19]. It was based on the original, unidimensional Ego-Resiliency Scale (ER89) developed by Block and Kremen [6]. The Polish adaptation of Block’s scale consists of 12 questions divided into two subscales: optimal regulation (OR) and openness to life experiences (OL). It has been proven, that OR is closely linked to stability and OL is linked to flexibility, which are the two components of ER as indicated in the Section 1 [19]. Responses were given on a 4-point Likert scale, ranging from 1 to 4. The OR subscale contains four questions (from 6 to 9), while the remaining questions (1–5; 10–12) relate to the OL subscale. The possible scores range from 12 to 48 for the whole test. The minimum score on the OL subscale is 8 and the maximum score is 32, while on the OR subscale is 4 and 16, respectively. Higher scores indicate higher intensity of ego-resiliency. The reliability of the adapted scale measured with Cronbach’s alpha coefficient was 0.822 for the whole test, 0.784 for the OR subscale and 0.768 for the OL subscale [19]. In this study, the Cronbach’s alpha coefficient for the whole test was 0.80.

#### 2.2.2. The Hospital Anxiety and Depression Scale (HADS)

The level of anxiety and depressive symptoms was measured using the Hospital Anxiety and Depression Scale (HADS) developed by Zigmond and Snaith [20]. The test consists of 14 questions divided into two subscales: an anxiety subscale (HADS-A) and a depression subscale (HADS-D). Each subscale contains 7 questions. Responses were given on a 4-point Likert scale, ranging from 0 to 3. The possible scores on each subscale range from 0 to 21. Higher scores on the HADS-A scale indicate greater severity of anxiety symptoms, whereas higher scores on the HADS-D scale signify greater severity of depressive symptoms. In validation studies among the Polish population, the scale was determined to be a reliable tool [21,22]. The Cronbach’s alpha coefficient for the whole test was 0.80.

#### 2.2.3. Satisfaction with Life Scale (SWLS)

The level of satisfaction with life was measured using the Polish adaptation of the Satisfaction with Life Scale (SWLS) [23]. The original tool was developed by Diener et al. [24]. The test consists of 5 questions. Responses were given on a 7-point Likert scale, ranging from 1 to 7. Higher scores signify greater satisfaction with life. The psychometric properties of the Polish version of SWLS were satisfactory and there the internal consistency (assessed by Cronbach’s alpha) was 0.86 [25]. The Cronbach’s alpha coefficient in our study was 0.87.

### 2.3. Socio-Demographic Variables

In our sociodemographic questionnaire, we collected data about age, gender, education and place of residence of our participants. Participants were obliged to tape in their number of years and to choose between three possible answers regarding their gender: male, female and other. Participants were also requested to provide information about their place of residence regarding the number of citizens. Educational status of our participants was classified as primary and vocational education (people who completed 8 years of primary school and 3 years of vocational school), secondary education (people who completed 3 years of high school or 4 years of technical school), and higher education (people who completed at least 3 years of university and obtained a university diploma).

### 2.4. The Subjective Assessment of Changes in Mental Condition Due to the Pandemic

In order to subjectively assess the mental condition of the respondents, subjects were asked “Have you noticed any change in your mental condition (stress, anxiety) caused by the occurrence of the pandemic?”. Participants rated their condition by choosing one of the three given responses: “Yes, my mental condition has deteriorated”; “Yes, my mental condition has improved”; or “No, I have not noticed any change in my mental condition”. Depending on their answers, respondents were divided into three groups: positive, negative, or no subjective change in their mental condition.

### 2.5. Data Analysis

The collected data was analysed using the computer programs Excel 2016 and Statistica version 13.3. The Cronbach’s alpha coefficients were determined for all measures used in the study. The Shapiro–Wilk test was used to assess the normality of the distributions. The Spearman’s rank correlation coefficient was used to measure correlations between ER, its components, and anxiety and depressive symptoms in the respondents. The Kruskal–Wallis test was used to compare the three answer groups. The level of statistical significance was set at α ≤ 0.05.

### 2.6. Ethics

The study was conducted in accordance with the Declaration of Helsinki. The university’s Bioethics Committee approved the study procedure. All participants provided informed consent. Respondents did not receive any reward for participating in our study.

## 3. Results

A total of 604 Polish volunteer participants took part in the study: 468 females (77.50%) and 136 males (22.50%). The average age of our participants was 28.95 ± 11.27 years. All responses given in the questionnaires were complete, valid, and included in the study. The sociodemographic characteristics of the study group are shown in Table 1.

The significant correlations between ER, depression, anxiety, and satisfaction with life were observed in the study group (Table 2). The study revealed a significant negative correlation between the intensity of the ER and the severity of depressive and anxiety symptoms. The same correlation was found in relation to OR subscale. Significant positive correlations at a moderate level were observed between the ego-resiliency components (OR and OL). The intensity of the ER also correlated positively with the intensity of satisfaction with life.

Table 3 presents differences in the results obtained for ER in general, OR subscale, OL subscale, and SWLS between the three groups of participants: those with positive subjective change, negative subjective change, and no subjective change in mental condition due to the pandemic.

The comparative analysis revealed statistically significant differences between these three groups in terms of ER and OR. The group with no change in mental condition was characterized by significantly higher resiliency. No significant differences in the results obtained for subscale OL were noted among the three categories of subjects. Similarly, significant differences were found between the three groups in terms of SWLS. The group with no change in mental condition was characterized by a significantly greater satisfaction with life.

## 4. Discussion

Our study revealed a significant negative correlation between the intensity of ER and the severity of depressive and anxiety symptoms: a result that has also been demonstrated by previous studies [2,26]. One of the mechanisms that allows ego-resilient individuals to avoid severe depressive and anxiety symptoms might be the ability to quickly and effectively restore their psychological balance (understood “as the state where an individual’s level of consistency and flexibility reconciles their perceived internal and external worlds” [27]) after facing adversity. Ego-resilient individuals use positive emotions to ‘bounce back’ from stressful situations. This phenomenon was described by the ‘broaden-and-build theory of positive emotions’ [28] and is linked to Block’s ER, as well as processual resiliency [1].

Moreover, ego-resilient individuals are characterized by positive emotionality and have the ability to induce and enhance their positive emotions through the use of humor, relaxation, and optimistic thinking [1,7,29]. As the broaden-and-build theory of positive emotions suggests, there is a close connection between positive emotionality and flexibility: when experiencing positive emotions, we are able to access a wider range of behaviors and ideas [1]. Hence ego-resilient individuals are able to see and consider a broadened spectrum of possible solutions when facing difficulties. Furthermore, ego-resilient individuals can successfully engage in building supportive social networks due to their ability to arouse positive emotions in people close to them [30]. Communal support and avoiding loneliness and isolation reduce the risk of depression and anxiety [31,32]. Lastly, high resiliency ensures higher tolerance towards negative emotions [2].

Our results suggest that ER seems to have an impact on the objective (lower levels of anxiety and depression among more ego-resilient individuals) as well as the subjective mental condition. The latter was assessed through an analysis of the responses given to the question, “Have you noticed any change in your mental condition caused by the occurrence of the pandemic?”, which was included in our survey. Those who did not report any changes in mental condition were characterized by higher ER.

In conclusion, ER’s impact on mental health may be crucial during crises. A meta-analysis of the prevalence of depression and anxiety among COVID-19 patients conducted by Deng et al. showed that 45% suffered from depression and 47% from anxiety [33]. Because of the high prevalence of anxiety and depression, numerous studies [3,26,34] have explored the impact of ER on people’s functioning during the pandemic. Some studies show that high ER may help to maintain psychological balance mainly due to its negative correlation with anxiety [3]. Nonetheless, we would like to point out that anxiety seems to play an important role in triggering mechanisms of ER [8]. In other words, the first reaction to external or internal stimuli is anxiety; this subsequently triggers the mechanisms of ER. which select one of two strategies: accommodation or assimilation. Emotions which activate ER traits promote flexibility in our personality system, and this allows us to access a broadened spectrum of coping mechanisms.

Another focus of our study was the correlation between life satisfaction and ER. Due to the fact that anxiety and depression correlate negatively with life satisfaction [15,35,36], we suspected that ego-resilient individuals might score higher on the Satisfaction With Life Scale due to lower prevalence of these disorders. In our study, we observed a positive correlation between life satisfaction and high levels of ER. It has been confirmed that ER is a mediator between negative aspects of psychological well-being (loneliness, hopelessness, and depression) and life satisfaction [37,38]. Moreover, ER might facilitate building supportive social networks, which reduce the feeling of loneliness [30]. Loneliness correlates negatively with life satisfaction itself [39,40]; but it is also linked to a higher level of hopelessness, which can lead to depression [37]. During the COVID-19 pandemic the lack of social contacts was the strongest predictor of life satisfaction. In other words, individuals who lacked social contact were less satisfied with their lives [41].

A study carried out by Ziarko et al. revealed a positive correlation between ER and coping strategies (problem- and emotion-oriented strategies) [38]. It has been proven that ER plays a mediating role between coping strategies and life satisfaction, especially towards emotion-coping strategies such as reframing, acceptance, and seeking emotional support [38]. Patients who evinced these coping strategies were characterized by higher level of ER and higher life satisfaction. Interestingly, ER correlated with one of the dysfunctional coping strategies, namely self-distraction. Individuals who adopted both active coping strategies and self-distraction had higher life satisfaction.

In our study, the group of respondents who did not report any subjective changes in mental condition were characterized by higher life satisfaction, which may indicate that experiencing life satisfaction depends more on our attitude and approach to life than on environmental changes.

As indicated earlier, ego-resilient individuals are characterized by a ‘positive emotionality’. Positive emotions can be helpful in achieving desired outcomes [42]; moreover, they predict an increase in resiliency [43]. On the one hand, individuals who often experience positive emotions are more satisfied with their lives due to the ability to enjoy themselves. On the other hand, positive emotions contribute towards life-satisfaction by helping develop the resiliency necessary for ‘bouncing back’ from negative emotions and adversity. A study conducted by Cohn et al. [43] indicated that the role of positive emotions is absolutely crucial in personal growth—especially in building resources such as resiliency and seeing a broadened spectrum of opportunities. Nevertheless, based on our results, it is not possible to determine the causality between high levels of ER and life satisfaction: in other words, we cannot predict if high levels of ER enhance life satisfaction or if higher life satisfaction helps to establish resiliency. Therefore, the aim of our considerations about the observable link between these two variables is only to make an attempt to explore some of the possible interplays between them.

## 5. Limitations

The present study has some limitations. We used the convenience sampling method, which cannot provide a random selection of a sample. The majority of our participants were female young adults (20–29 years old), as well as teenagers (16–19 years old) with secondary and higher education. Therefore, the results of our study are most relevant for young, well-educated (meaning secondary and higher education) women. Furthermore, our results and conclusions need to be treated with caution due to the young age of many of our respondents. Peoples’ personalities during teenage years and early adulthood tend to develop and change dynamically; this process might bias the results. Moreover, our study is a cross-sectional study and, therefore, it is impossible to establish causality between the variables studied. Our survey was carried out via social media networks, which means that only individuals with internet access and social media accounts were able to take part in our study. The question used to assess the subjective mental condition of the respondents was originally established by the authors of this study, which should be revised and improved in future studies. Furthermore, our assessment of well-being is based on the assumption that our respondents were able to assess it using their own judgement (in other words: subjective sense of well-being). We did not assess any objective factors (i.e., living conditions) of well-being.

## 6. Conclusions

Individuals with a high level of ego-resiliency might experience a lower intensity of anxiety and depressive symptoms during crises such as the COVID-19 pandemic. Furthermore, individuals with a high level of ego-resiliency might exhibit a higher level of life satisfaction.

## Figures and Tables

**Table 1 ijerph-19-10364-t001:** Sociodemographic characteristics of the study group and descriptive statistics.

Variables	Frequency (n = 604)	Percentage (%)
Age		
16–19	60	10.00
20–29	360	60.00
30–39	51	8.00
40–49	87	14.50
50+	46	7.50
Gender		
Female	468	77.50
Male	136	22.50
Education		
Primary and vocational	14	2.00
Secondary	373	62.00
Higher	217	36.00
Place of residence		
Village	193	32.00
City with ≤100,000 inhabitants	190	31.00
City with >100,000 inhabitants	221	37.00
Psychometric	Mean (SD)	Median (IQR)
Ego-resiliency	34.64 (5.96)	35.00 (31.00–39.00)
OR	22.56 (4.21)	23.00 (20.00–25.00)
OL	12.08 (2.63)	12.00 (10.00–14.00)
Anxiety	8.94 (3.70)	9.00 (6.00–11.00)
Depression	5.51 (3.39)	5.00 (3.00–8.00)
Satisfaction with life	21.06 (6.46)	21.00 (17.00–26.00)

Notes: OR—Optimal Regulation; OL—Openness to life experience; SD—standard deviation; IQR—Interquartile range.

**Table 2 ijerph-19-10364-t002:** Associations between ER, its components, anxiety and depressive symptoms, and satisfaction with life in the entire study group.

	ER	OR	OL	Anxiety	Depression	Satisfaction with Life
ER	1.000	0.909 *	0.781 *	−0.246 *	−0.267 *	0.358 *
OR		1.000	0.472 *	−0.304 *	−0.284 *	0.405 *
OL			1.000	−0.073	−0.150 *	0.158 *
Anxiety				1.000	0.444 *	−0.326 *
Depression					1.000	−0.371 *
Satisfaction with life						1.000

Notes: The Spearman’s rank correlation coefficient was used to assess correlations between variables; values are significant at * *p* < 0.05.

**Table 3 ijerph-19-10364-t003:** Differences between changes in mental condition in the entire study group.

Variables	Positive Change	Negative Change	No Change	Kruskal–Wallis Test
	Me (IQR)	Me (IQR)	Me (IQR)	*p*
ER	36.00 (32.00–38.00)	34.00 (29.00–38.00)	36.00 (33.00–40.00)	0.000 *
OR	23.00 (20.00–25.00)	22.00 (19.00–25.00)	24.00 (21.00–26.00)	0.000 *
OL	13.00 (10.00–14.00)	12.00 (10.00–14.00)	12.50 (11.00–14.00)	0.343
Anxiety	9.50 (7.00–11.00)	10.00 (7.00–12.00)	7.00 (5.00–9.00)	0.000 *
Depression	4.00 (2.00–6.00)	6.00 (4.00–9.00)	4.00 (2.00–5.00)	0.000 *
Satisfaction with life	23.00 (19.00–26.00)	20.00 (16.00–25.00)	23.00 (19.00–28.00)	0.000 *

Notes: values are significant at * *p* < 0.05.

## Data Availability

Data supporting reported results are available on request from the corresponding author.

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
