# Peer review of "Depressive and Anxiety Symptoms and Their Relationships with Ego-Resiliency and Life Satisfaction among Well-Educated, Young Polish Citizens during the COVID-19 Pandemic"

_ijerph, 2022, doi:10.3390/ijerph191610364_

Round 1

Reviewer 1 Report

Manuscript ID ijerph-1806610

Title: Depressive and anxiety symptoms and their relationships with ego-resiliency and life satisfaction among Polish citizens during the Covid-19 pandemic.

Dear authors,

Your work is very interesting, and although it is only descriptive results, I believe it can be useful for the research community. However, you should make some changes to the manuscript to make it more readable and understandable.

Manuscript ID ijerph-1806610

Title: Depressive and anxiety symptoms and their relationships with ego-resiliency and life satisfaction among Polish citizens during the Covid-19 pandemic.

Dear authors,

Your work is very interesting, and although it is only descriptive results, I believe it can be useful for the research community. However, you should make some changes to the manuscript to make it more readable and understandable:

-          Introduction: References in text are incorrect (i.e: [1][2][3], should be [1,3]). Please, check this throughout the manuscript.

-          Introduction -first paragraph-: There is a lack of fluence between ideas. You have to write short and to the point. I suggest you change the first sentence for: “Resiliency is one of the crucial coping mechanisms which promotes healthy and beneficial behaviors during crises [1-3]” Then state a clear definition of resiliency (only one, you have put two, one in line 29 and other in line 33), and finally ER definition.

-          Line 36: Add a short description of flexibility and stability.

-          Line 42: Use OR.

-          Lines 42-45: It is not very clear why do you describe the ER89, OR, and OL if after this paragraph, those terms don’t appear again. You should connect OR an OL with “ego-resilient” individuals, or eliminate this information.

-          Lines 59-64: In my opinion, you give a lot of information about cytokines and depression, an association that is not your aim of study. You have to try to connect ego-resiliency with depression, but without physiological concerns.

-          2.1 Data and participants: You should change the name of this subsection to “Design and participants”. In it, you should state clearly “Cross-sectional study” and then, the Google Form description. Related to participants, you have to include inclusion and exclusion criteria.

-          2.1 Data and participants: In addition, you cannot include a table in methods. All information from lines 97-109 is results information.  

-          2.2.1. Ego-resiliency Scale (ER89-R12): Maybe you can add here the introduction information about this questionnaire. Don’t repeat the abbreviations again. Total score? Cut-off scores? Higher scores are better or worse? Please add some information in this regard.

-          HADS and ER89-R12 are validated in polish population? Please, add this information.

-          Line 144: And satisfaction with life?

-          Results: This section should start with a short paragraph about results of table 1. 

-          In general, the results are difficult to read. Avoid writing "in this table ..." all the time. It is more concrete and direct to write: Spearman correlation results show that people with more ER seem to have higher life satisfaction and lower anxiety and depression (table 4).

-          Table 1: It is not clear why you have chosen these age cut-off points. Nor is its clear what tertiary education means. It is necessary to add a sub-section in methodology explaining the socio-demographic variables used.

-          Table 2: Join this table to table 1. Table 2 alone is not very informative. Also, you have used spearman correlation and Kruskal Wallis so I understand that the distribution of the variables is not normal. In this case, quantitative variables should be described with median (interquartile range), without using the +- symbols, as this is incorrect.

-          Table 3 footnotes: add the test used to obtain p-values.

-          Table 4: Why do the variables anxiety and depression not appear?. In addition, variables should be described with: median (interquartile range).

-          Lines 221-223: I think you cannot derive this "conclusion" from your analyses, as they are cross-sectional and you cannot establish causality. You should be more careful in the message you send.

-          Lines 225-229: This is a very isolated paragraph. You should try to make the reading flow smoothly. Sometimes it becomes too difficult.

-          Line 244: Sometimes you use ER and sometimes the full name, unify.

-          In my opinion, throughout the discussion it is taken for granted that RE is the one that influences the rest of the variables studied, but you should not forget that you have carried out a cross-sectional study, in which it is impossible to know which variable influences another. In other words, we cannot know whether better life satisfaction makes us have better RD or better RD makes us have better life satisfaction. I suggest you review the discussion with this comment in mind.

-          Line 263: It is very curious that more than half of the participants are between 16-19 years old and there is such a high rate of tertiary education. It is very necessary that you include a good and clear definition of socio-demographic variables in the methodology.

-          Limitations: Add the impossibility of establishing causality between the variables studied as the results obtained come from a cross-sectional study.

-          I am not so sure that conclusion 3 follows from your results. Please write the conclusions instead of listing them.

Thank you.

Reviewer 2 Report

Dear Authors.

This paper is very well proven statistically. Yet, the foundations of determinants which were the subject of evaluating relations are, at least, disputable.

Line 14 – the age of 16 is it an adult?

Lines 22-23 This statement is unjustified in the context of Your conclusions 

Line 35 “Healthy personality traits”, what constitutes them and differ from “unhealthy” ones….

Line 61 Have You measured the participants’ level of cytokines?

Lines 63 – 66. “good point” – according to the majority of Your research sample, it would be very significant to compare the stress level when the mobile phones are taken from research participants for (f.e.) a month. (com. Table 2). Are You convinced the young people (16-19 years of age) are able to distinguish the depression from other similar factors (f.e. sadness, a lack of comfort)

79 – “our lives” You mean whose? F.E. mine not…..

Line 80-82 This is a set of unjustified postulates. There are more of them in Your study….

Lines 81-82 – You did not prove it?

Line 83- what where the “satisfaction” criteria in Your study?

Line 85- Did You questioned a single person, which has been satisfied with a pandemic (except bankers and the big pharma)?

Line 102. Referring to Costa and McCrae BIG5 model, personality traits can be measured precisely under one crucial condition - the age over 30. Hence, 68% of Your research sample is biased.

[com. f.e. DOI: 10.7341/20181418

https://doi.org/10.1016/j.paid.2020.109816
 https://doi.org/10.1037/0021-9010.82.1.30].

In that context, it is not a good idea to implement the induction I’m afraid.

Line 220 - since when the changes are objective? If not, what is the reason to differ?

Line 265 What does it mean “well-educated women” and who is responsible for such an evaluation (who is well and not). What constitutes “well education”?

Line 270. Conclusions.

The higher the better? What about the Yerkes-Dodson Laws?

What is a novelty in this study?

I have not indicated any statements which were not described (proven) earlier – this study seems to me “a new wine in the old bottle”

The English language should be corrected. There are too many common language examples. You should be also very careful in the context of postulates.

Round 2

Reviewer 1 Report

Manuscript ID ijerph-1806610

Title: Depressive and anxiety symptoms and their relationships with ego-resiliency and life satisfaction among Polish citizens during the Covid-19 pandemic.

Dear authors,

Thank you for taking my comments into account and for responding to most of them.

There are still some minor issues that you should address:

-          Introduction: In my past comments, I made a mistake. In the references [1][2][3], I meant to say that they should be written like this [1-3], as they are consecutive numbers. That is why a hyphen is included. Revise throughout the manuscript.

-          Lines 154-162: This new paragraph should be in methodology. Maybe, you can add a “other variables” subsection in methodology and include in it this information. In fact, it is one of the comments I made to you in the previous review: “It is necessary to add a sub-section in methodology explaining the socio-demographic variables used”

-           Line 162: higher  =   tertiary? Use the same terms as in the tables.

-          Table 1: When a quantitative variable does not follow a normal distribution, it is usually shown in tables only with median and interquartile range. If you also want to show mean and standard deviation, it is up to you. But, you should display both sets as "Median (IQR)" and "Mean (SD)" . “Median SD” example: 34.64 (5.96). The use of the symbols + and - is incorrect. Also, you should use only two decimal places. “Median (IQR) example: 35 (15-25) . The IQR is represented as (percentile 25-percentile 75)

-          Table 1: If you write Percentage (%) in the caption, you can delete the % symbol beside the data.

-          Now, table 3 is table 2. Revise the tables’ names

-          Table 4: Use only one column per category. Represent the results as Median (IQR). Remember that IQR is (p25-p75)

-          Table 4: What is H? (Kruskal wallis). I think that you can delete this column and leave only the p-values. In addition, is repetitive to use * when a p-value is significant.

-          Limitations: Add the impossibility of establishing causality between the variables studied as the results obtained come from a cross-sectional study.

Thank you.

Reviewer 2 Report

Attached
